# Boosting gradient-based optimizers for asynchronous parallelism

**Shuai Li, Yi Ren, Dongchang Xu, Lin Guo, Hang Xiang, Di Zhang, Jinhui Li**
Alibaba Inc.
Beijing, China
{voolc.li,hengrui.ry,dongchang.xu,lin.gl}@alibaba-inc.com
{xingzhi.xh,di.zhangd,jinhui.li}@alibaba-inc.com

## Abstract

Stochastic gradient descent methods have been broadly used in training deep neural network models. However, the classic approaches may suffer from gradient delay and thus perturb the training under asynchronous parallelism. In this paper, we present an approach tackling this challenge by adaptively adjusting the size of each optimization step. We demonstrate that our approach significantly boost SGD, AdaGrad and Momentum optimizers for two very different tasks: image classification and click through rate prediction.

## 1 Introduction

Deep learning techniques have demonstrated great potential in many industrial applications, e.g., computer vision (He et al., 2016) , speech recognition (Sak et al., 2014), natural language processing (Gehring et al., 2017), and computational advertising (Wang et al., 2017), etc. The rise of deep neural networks requires massive data and is accelerated by the modern advance in computing technologies. Parallel and distributed computation has greatly benefited deep learning implementations (Dean et al., 2012) by drastically reducing training times, such that larger amount of data or more demanding algorithms can be explored within an acceptable time cost.

Stochastic Gradient Descent optimizing(Bottou, 1998) approach has been proven extremely useful for solving large-scale machine learning problems, due to its simplicity and robustness. However, subtle work is needed to tune the hyper-parameters to train the model efficiently, i.e., achieving a better convergence within fewer iterations. Extensive research efforts have been made to develop more efficient optimizers. For example, Adaptive gradient (AdaGrad(Duchi et al., 2011)) a simple but popular method. With an adaptively decaying learning rate, AdaGrad is suitable for sparse data and asynchronous parallel training.

Many works have been conducted on extending traditional optimizing algorithms to parallel and distributed deep learning, especially on Asynchronous Stochastic Gradient Descent (ASGD) optimization (McMahan & Streeter, 2014; Liu et al., 2015; Zheng et al., 2016). ASGD allows each local worker to work independently, i.e., computing the gradient over its own mini-batch of data, adding the gradient to the global model, and then pulling the updated global model back for the next step of iteration. Without the barrier of synchronization among workers as classic synchronous Stochastic Gradient Descent, each worker continues its training process immediately after communicating with the global model. As a result, large-scale parallel training is significantly speeded up.

For asynchronous parallelism under parameter server framework, a local worker computes the gradient $g_t$ based on the global model status (denoted by $w_t$) at global step $t$. Before $g_t$ is applied to update the global model, the global model has already been updated to $w_{t+\tau}$ by the gradients from other workers. Therefore, $g_t$ becomes delayed for the global model. Updating the global model with delayed gradients is not always mathematically safe, and may perturb the training trajectory. Moreover, this perturbation becomes more severe as the parallelism scales up.

In this paper, we propose an optimizing approach tackling the challenges from the gradient delay and training perturbations. For each trainable parameter of the neural network model, our approach utilizes it's relative increment between $t + \tau$ and $t$ to adjust the learning rate adaptively, and thus

the perturbation is relieved. We introduced this adaptive mechanism to boost SGD, AdaGrad and Momentum optimizers and conducted the experiments in two very different scenarios: image classification and click through rate (CTR (McMahan et al., 2013)) prediction. Results show that our approaches outperform all the original optimizers.

## 2 METHODOLOGY

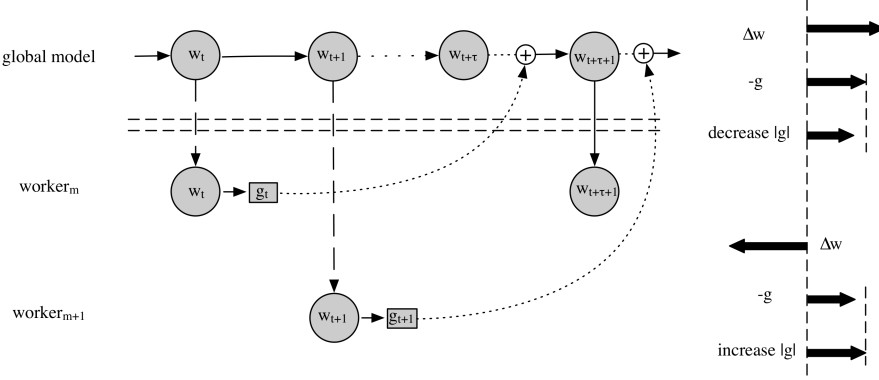

Figure 1: Asgd training process.

With respect to $g_t$, the weight relative increment between $w_{t_\tau}$ and $w_t$ is defined as

$$\Delta w_t = w_{t+\tau} - w_t. \tag{1}$$

As illustrated in Fig. 1, $work_m$ receives global model $w_t$, and then passes the calculated gradient $G_t$ back to the global model which has already been updated by other workers to $w_{t+\tau}$ during the same time. For each trainable parameter, we use the relative direction between $g_t$ and $\Delta w_t$ to adjust the learning rate adaptively when updating $w_{t+\tau}$. When $-g_t$ and $\Delta w_t$ have the same sign, it implies the model's parameter has been updated towards the same direction as $-g_t$ during the past $\tau$ steps by the other works. Under this condition, the optimizer needs to be more conservative about the magnitude of the new update. For this concern, the learning rate is tuned down to weaken the delay gradient signal. On the other hand, if $-g_t$ has the opposite sign against $\Delta w_t$, the gradient signal may bring new information of exploration to the other direction. The optimizer increases learning rate to encourage this exploration.

The algorithms of boosted SGD, AdaGrad and momentum optimizers with our approach are described in algorithms 1, 2 and 3, respectively. The product of $g$ and $\Delta w$, is normalized into the regime between $-1$ and $1$, which is necessary to stabilize the algorithm. Two ways of normalizations are implemented. One way is simply picking up the sign, i.e.,

$$normalize\_factor = sign(g \cdot \Delta w). \tag{2}$$

This produces an adorable performance for AdaGrad. SGD and momentum requires a more delicate adjustment, i.e.,

$$normalize\_factor = g \cdot \Delta w/reduce\_max(|g \cdot \Delta w|), \tag{3}$$

where $reduce\_max$ returns the maximum for all model parameters. A hyper parameter $\lambda$ is introduced to control the magnitude of adjustment. We found that $\lambda = 0.3 \sim 0.4$ produces best in our experiments.

## 3 EXPERIMENTS

We demonstrate the effectiveness of our approach for two tasks: image classification and CTR prediction. For image classification, we trained AlexNet(Krizhevsky et al., 2012) on cifar10, a

---

**Algorithm 1** boosted-SGD

---
1: $normalize\_factor = g \cdot \Delta w / reduce\_max(|g \cdot \Delta w|)$
2: $g \leftarrow (1 - \lambda * normalize\_factor) * g$
3: $w \leftarrow w - \eta * g$

---

**Algorithm 2** boosted-AdaGrad

---
1: $normailze\_factor = sign(g \cdot \Delta w)$
2: $s \leftarrow s + (1 + \lambda * normalize\_factor) * g^2$
3: $w \leftarrow w - \eta * g / sqrt(s + \epsilon)$

---

**Algorithm 3** boosted-momentum

---
1: $normalize\_factor = g \cdot \Delta w / reduce\_max(|g \cdot \Delta w|)$
2: $g \leftarrow (1 - \lambda * normaliz\_factor) * g$
3: $m \leftarrow \beta * m + \eta * g$
4: $w \leftarrow w - m$

---

public data set which consists of 32 x 32 color images drawn from 10 and 100 classes split into 50,000 train and 10,000 test images. For CTR prediction, we trained a deep neural network model (Cheng et al., 2016), with 5 fully connected layers, on a data set collected from the online adverting platform of our company. The training and test sets contain 4 billion and 800 million instances, respectively, with 10 billion unique feature ids in total. For each model, we firstly generate a set of randomized initial parameters to be leveraged by all the related experiments.

For each original optimizer and each task, we tuned the hyper-parameters to achieve the best performance. The boosted version of optimizer was then applied with the same hyper-parameters to train the same model. For each experiment configuration, we run 5 times to report the average numbers. As presented in tables 1 and 2, our approach significantly enhance the performance of all the three optimizers, i.e., SGD, AdaGrad and momentum, for the both tasks. Note that the image classification and the CTR prediction are very different tasks. The success of our approach for the two tasks manifests that our approach has the potential to be applied over a wide range of scenarios.

Table 1: Performance for CTR prediction. Relative AUC gain of boosted optimizers with respect to original ones.

| parallel num | boosted-sgd | boosted-moment | boosted-adagrad |
|---|---|---|---|
| 100 | +0.012% | +0.01% | +0.012% |
| 200 | +0.028% | +0.045% | +0.051% |

Table 2: Performance for Cifar10. classification accuracy of boosted optimizers with respect to original ones.

| parallel num | sgd | boosted | moment | boosted | adagrad | boosted |
|---|---|---|---|---|---|---|
| 30 | 82.91% | +0.43% | 83.43% | +0.2% | 83.06% | +0.25% |
| 60 | 82.48% | +0.56% | 82.67% | +0.25% | 82.37% | +0.46% |

## 4 CONCLUSION

In this work, we proposed an approach to enhance large-scale asynchronous distributed optimization for deep neural networks. Gradient delay and training perturbations are relieved by adaptively adjusting the size of each optimization step. The effectiveness of our approach was demonstrated by successfully boosting SGD, AdaGrad and Momentum optimizers for two very different tasks: image classification and CTR prediction. For future work, we will further explore effective optimizations under large-scale parallelism for industrial-level implementations.

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
