# OpenReview forum: "Boosting Gradient-based Optimizers for Asynchronous Parallelism"
_ICLR.cc/2018/Workshop — Reject_

### Official Review · AnonReviewer2 · 2018-03-04
**Lack of convergence discussion**

**Rating:** 4
**Confidence:** 2

**Review:**

This paper introduces a simple method to adjust the size of gradients for alleviating the perturbation problem in ASGD. This method emphasizes the gradient which has different phenomenon as compared with other gradients. As a result, experimental results show that their approach boost performances of asynchronous stochastic optimizers.

The concern of this paper is the lack of the discussion of convergences. I could not understand whether this method is guaranteed to converge to some local minimum or oscillates around the local minimum. Even if this method could not have a theoretical guarantee, I think the discussion about the convergence is needed either theoretically or experimentally.

* Pros
- This paper proposes a simple method to alleviate the perturbation problem in the asynchronous SGD training due to gradient delay.

* Cons
- This paper lacks the convergence proof of this method and related discussions.

---

### Official Review · AnonReviewer3 · 2018-03-10
**Promising ideas, unconvincing evaluations**

**Rating:** 5
**Confidence:** 4

**Review:**

Summary
Adapting learning rates to account for delayed gradients in Asynchronous update methods. The idea is applied to three different optimizers for two different tasks and shown to improve over raw baseline for those methods.

Pros
- The idea is interesting, simple and relevant. It is easy to scale up asynchronous methods in cloud like environments and having ways for them to be more effective is quite useful.
- The evaluation is applied to two fairly different models and consistently works across each of the optimizers

Cons
- The work needed for each of the optimizers is different, hence one would need to do the same for every new optimizer. The good thing is that the basic ideas are simple enough and similar.
- Eval only compares a single baseline (no adaptation) to their boosting. No comparisons even with papers cited that offer solutions to delays in gradients e.g. DC-ASGD.

Minor clarification. Table #1 and Table #2 have parallel num that appears to number of parallel workers/updates. Will be great to clarify that.

---

### Official Review · AnonReviewer1 · 2018-03-13
**Possibly worthwhile exploration of adaptive step size based SGD.**

**Rating:** 5
**Confidence:** 4

**Review:**

The paper presents a reasonable idea for adapting the step size during SGD. The others relate their proposal to existing adaptive step size methods such as Adagrad. The weakness of the work is the messy and very often ungrammatical writing style. In my view this puts it slightly below the acceptance threshold. It should be proof-read by a native speaker of english. "This produces an adorable performance for Adagrad"??

---

### Decision · Program_Chairs · 2018-03-20
**ICLR 2018 Workshop Acceptance Decision**

**Decision:**

Reject

**Comment:**

Based on the reviews, this paper has not been accepted for presentation at the ICLR workshop. However, the conversation and updates can continue to appear here on OpenReview.